# Enhancing Memory for Relationship Actions by Transcranial Direct Current Stimulation of the Superior Temporal Sulcus

**DOI:** 10.3390/brainsci10080497

**Published:** 2020-07-29

**Authors:** Hipólito Marrero, Sara Nila Yagual, Enrique García-Marco, Elena Gámez, David Beltrán, Jose Miguel Díaz, Mabel Urrutia

**Affiliations:** 1Departamento de Psicología Cognitiva, Social y organizacional, Universidad de La Laguna, 38071 San Cristóbal de La Laguna, Spain; pssarukita@hotmail.com (S.N.Y.); egarciam.psicologia@gmail.com (E.G.-M.); egamez@ull.es (E.G.); dbeltran@ull.es (D.B.); jmdiaz@ull.es (J.M.D.); 2Instituto Universitario de Neurociencias de la Universidad de La Laguna, 38071 San Cristóbal de La Laguna, Spain; maurrutia@udec.cl; 3Facultad de Ciencias Sociales y de la Salud, Universidad Estatal Península de Santa Elena (UPSE), La Libertad Santa Elena, Provincia de Santa Elena EC240250, Ecuador; 4Facultad de Ciencias de La Salud, Universidad Europea de Canarias, 38300 La Orotava, Spain; 5Facultad de Educación, Universidad de Concepción, Victor Lamas, Concepción 1290, Chile

**Keywords:** approach/avoidance intentionality, relationship action-sentences, tDCS, Memory, Superior Temporal Sulcus

## Abstract

We examine the effect of transcranial direct current stimulation (tDCS) of right superior temporal sulcus (rSTS) in memorization of approach/avoidance relationship-action sentences; for example, “Alejandro accepted/rejected Marta in his group.” Sixty-five university students participated in a tDCS study, in which a between-subjects design was adopted. Sixty-four participants were also given the behavioral approach system (BAS) and behavioral inhibition system (BIS) scales. Participants were subjected to 20 min of stimulation: anodal (N = 24), cathodal (N = 21), or sham (N = 20); subsequently, they were given a list of 40 sentences (half approach and half avoidance) and told to try to memorize them. Finally, they performed a changed/same memory task (half the sentences were the “same” and half were “changed”). Previously, we had examined performance in the memory task without tDCS with another group of participants (N = 20). We found that anodal stimulation improved *d’* index of discriminability (hits-false alarms) compared to sham and cathodal conditions for both approach and avoidance sentences. Moreover, the comparison between anodal and task-alone performance showed that stimulation improved *d’* index of approach sentences more, as task-alone performance showed better discrimination for avoidance than for approach. Likewise, we explored a potential modulation of tDCS effect by (BAS) and (BIS) traits. We found that *d’* index improvement in anodal stimulation condition only benefited low BAS and low BIS participants. Implications of these results are discussed in the context of rSTS function in encoding and memorizing verbally described intentional relationship-actions and the role of individual differences on modulating tDCS effect.

## 1. Introduction

Intentionality is a basic component of understanding the minds and behaviors of others. In this regard, the temporal lobe (anterior temporal lobe, superior temporal sulcus, middle and superior temporal gyrus) and also the precuneus and temporo-parietal junction constitute a “mentalizing” network [1,2,3] that encodes intentionality. It is relevant to distinguish between representation of intentions as mental states not associated with current actions and representation of intentions and goals that are inherent in perceived actions. The latter involves a neural system particularly associated with the superior temporal sulcus (STS) and is recruited for action understanding [4]. Moreover, activation of this mentalizing network to process social information is usually stronger in the right hemisphere [5,6].

Within the mentalizing network, the superior temporal sulcus (STS) and brain areas around it have been shown to be particularly involved in processing communicative intention for interactions by means of gaze (direct vs. averted) in social perception [7,8,9,10,11] and mutual liking [12]. It has also been shown that approach intentionality causes greater activation of posterior right superior temporal sulcus (rSTS) than avoidance. In a functional magnetic resonance imaging (fMRI) study [13], brain activation in response to a stranger initiating or avoiding social interaction was measured. Participants viewed an animated character approaching down a virtual hallway, who shifted his gaze either toward or away from the participant. Mutual gaze (approach) caused a greater activation in this brain region than averted gaze (avoidance). These studies usually focused on demonstrating that STS is responsive for action intentionality and social contexts, and not just for the more physical aspects of actions [7,8]. However, it could be that the STS is a brain area specifically recruited for processing intentionality for relationships.

Whereas social perception of approach/avoidance intentionality activates posterior aspects of rSTS, several studies have supported that more abstract and conceptual processing of relationship intentionality also recruits more anterior to middle aspects of rSTS. For example, Ross & Olson [14] (see also Tavares, Lawrence, & Barnard [15]), using a version of the Heider and Simmel animation task in a fMRI study, reported activation of more anterior aspects of rSTS when participants judged “friendship” from simple geometric shape interactions. Similarly, Gobbini, Koralek, Bryan, Montgomery, & Haxby [4] reported activation along the full length of rSTS when participants observe Heider and Simmel animations and made social intentional judgements of interactions.

Likewise, more anterior STS areas have been shown to be particularly active in processing scenes of social relationship interactions. In this line of research, significant activation of the anterior/middle STS has been reported specific to viewing video clips of relationship interactions [16,17] and in a verbal “theory of mind” task involving interactions in social relationship contexts. According to Iacoboni et al. [16], activation of more anterior aspects of the STS could represent the process of giving a social relational meaning to individual actions. This is in accordance with the role of the so-called Anterior Temporal Lobe (ATL), which includes more anterior aspects of the STS, in semantic processing of information and social cognition [6,14,18,19]. In particular, more anterior aspects of the STS and Superior Temporal Gyrus contribute more to abstract concepts and verbal semantic processing [20].

Beyond action observation, language describes how individuals interact with other people by means of social actions that conceptually involve approach “pro stimulus” and avoidance “against stimulus” intentionality [21,22]. For example, “Alejandro accepted/rejected Marta in his group.” Approach and avoidance would constitute a semantic frame or category to be systematically encoded for understanding this type of actions, as representing individual’s intentional direction towards other people has an adaptive role. Thus, if approach/avoidance gives meaning to relationship actions, we could expect activation of more anterior aspects of STS to process them in social relationship actions.

In a previous study [21], the hypothesis that understanding verbal expressions of others’ social actions would activate self-experienced approach/avoidance brain representations was tested. The electrophysiological activity of participants was recorded, while they were reading approach/avoidance action sentences from a character toward a target—a thing/a person (e.g., “Petra accepted/rejected Ramón in her group”/“Petra accepted/rejected the receipt of the bank”). Brain potentials were measured time-locked to the target word, namely the object of the sentence. This study found different event-related potentials ERPs for things and persons. For persons, a posterior negative ERP at the 545–750 ms time window emerged, with a left frontal distribution more negative to approach than to avoidance sentences.

In order to examine the brain areas involved in processing approach/avoidance interpersonal intentionality, we reanalyzed ERP data from Marrero, Urrutia, Beltrán, Gámez, Díaz [21] in the time window response for persons [23]. To estimate likely intracranial generators of the topographical approach-avoidance differences, we used the local auto-regressive average (LAURA) inverse solution approach [24]. As shown in Figure 1, we found that the approach/avoidance difference recruited the anterior to the middle right temporal brain area around rSTS (BA22, Talairach coordinates: x = 63, y = −16, z = 2) associated with greater activation for approach than for avoidance.

Previous research has supported that anterior to middle aspects of the STS and ATL are recruited for semantic processing of social concepts. In this regard, our source estimation analysis suggests that semantic processing of intentionality of approach/avoidance in relationship actions recruits rSTS. Thus, we hypothesize that middle aspects of rSTS will be involved in memorization of relationship actions.

In this paper, we examine, for the first time, the role of rSTS in memorizing intentional relationship actions. With this aim, we examine the effect of STS-targeted transcranial direct current stimulation (tDCS) on processing of relationship-action sentences quantified through recognition memory accuracy. We employ linguistic material: approach and avoidance sentences; for example: “Alejandro accepted/rejected Marta in his group.” We predict that tDCS stimulation on the rSTS will improve semantic encoding of relationship action-sentences, more strongly in approach sentences. According to Logan [25], encoding of and attention to stimuli are closely related processes. Thus, more processing resources, plausibly furnished by tDCS, would be associated with a deeper encoding of relationship actions (strongly for approach), and, in turn, a deeper encoding would be associated with better memorization.

The memory task consisted of a same-different task, where characters’ proper names appeared either in the original roles or with their roles exchanged in the sentence (the subject as the object and the object as the subject of the action), and participants had to judge whether the displayed sentence was “changed.” We used discriminability *d’* index as the dependent measure in accordance with signal detection theory [26]. We predicted that anodal stimulation would improve discriminability in the memory task, more strongly for approach than for avoidance.

### Moderation of tDCS Effect by Approach and Avoidance Personality Traits

We also explore whether or not an effect of tDCS on discriminability in our memory task is moderated by approach and avoidance traits—behavioral approach system (BAS) and behavioral inhibition system (BIS) [27]. Previous research has found the low approach trait benefitting from anodal stimulation [28,29].

As stimulation is induced in a temporal area (no executive functions are involved as in Metuki, Sela, & Lavidor [28]), we consider that the moderator effect of trait could be exerted by affecting attention allocation in possibly two different ways. One way is by a motivational bias. In this case, we expected that greater cognitive resources furnished by tDCS would be used more on processing approach sentences than avoidance by high BAS participants, whereas high BIS participants would use more resources processing avoidance sentences. Thus, we predicted greater tDCS effect on discrimination in approach than in avoidance sentences for high BAS trait, and in avoidance than in approach sentences in high BIS trait.

The other way is related to a deficit in attention allocation. Previous research has suggested that high approach trait is associated with less concentration, more distractibility, and less attentional narrowed focus on a given task [30,31,32]. In the case of avoidance trait, previous research has clearly showed that fearfulness and anxiety disturb the capacity for allocation of attentional resources to a particular task (see Eysenck, Derakshan, Santos, & Calvo [33]). Thus, we would expect that high BAS and high BIS traits participants would be less able to take advantage of additional processing resources plausibly furnished by anodal tDCS for an effortful memory task. Thus, we predict poorer discrimination both in approach and avoidance sentences in high in contrast to low BAS and BIS participants.

## 2. Study 1—Memorization of Approach/Avoidance Sentences

Previous to the tDCS study, we examined memorization performance in the experimental task performed alone. Task-alone performance shows the pattern of memorization performance for approach and avoidance sentences without the influence of tDCS and could provide us with a better understanding of tDCS effects. Committee of Ethics of Research and of Animal Welfare of University of La Laguna approved this study on 22 December 2017: (CEIBA 2017-0272).

### 2.1. Method

#### 2.1.1. Participants

Twenty undergraduate students (15 females, mean age = 20.5, SD = 1.43) from the University of La Laguna (La Laguna, Spain) participated voluntarily in the experiment in exchange for course credits.

#### 2.1.2. Stimuli

We selected a pool of approach and avoidance sentences from Marrero et al. [21]. Each sentence had different character names and action verbs to facilitate discrimination between them. One half of proper names were female, and the other half were male names. Verbs were in the past tense, indicating that actions had taken place, to highlight readiness for action encoding. In order to control for the influence of the proper names, proper names appeared both in approach and avoidance versions of sentences, and two lists of stimuli were elaborated. Each list had 40 sentences (20 approach and 20 avoidance sentences). In Table 1, sentence examples are shown in the different versions for the memorization phase and the subsequent memory task. In the memory task, participants had to judge whether the displayed sentence was the “same” or “changed.”

#### 2.1.3. Design

A 2 × 2 factorial design was used, with direction (approach and avoidance) and sentence (same vs. changed) as within-subjects factors. The dependent variable was performance in the memory task, measured by the percentage of correct responses, and *d’* index mean scores of discriminability signal-noise according to signal detection theory.

#### 2.1.4. Procedure

Participants were told that the task consisted of reading sentences that described relationship actions. Their goal was to memorize both the concrete action carried out and the names of the characters involved in the action and their role, since later on, there would be a same-changed memory task where the characters’ proper names could appear either with the same role or with their roles exchanged. They were also told that they would receive feedback on the percentage of correct responses they reached in the memory task. Likewise, they were told that after the memorization phase, they would have to perform a numerical task. The numerical task was aimed at avoiding rehearsal prior to memory task.

In the memorization phase, participants read sentences while seated in front of a computer screen. Each sentence presentation started with a cross point displayed in the middle of the screen for 750 ms. After an interval of 150 ms, one sentence was displayed. Sentence presentation was segmented (six segments); for example, “Pedro/bloqueó/a/Rosa/en el/Whatsapp” (“Pedro/blocked/Rosa/on/Whatsapp.”) Each segment was displayed for 300 ms with an interval of 150 ms between them. After display, the sentence remained on the screen for 1000 ms, and then a new sentence was displayed. Participants were given 40 sentences (20 approach and 20 avoidance). They were randomly assigned to one of the sets of sentences resulting from the counterbalance of the experimental conditions. Sentences were randomly presented in each of the counterbalancing sets.

Subsequently, a numerical task was displayed for around 2 min, and participants were told to respond whether numbers ranging from two to five figures were even or odd by pressing either the P key or the Q key on the keyboard, respectively. Finally, the memory task was given. Participants were presented with the list of sentences, sentence by sentence. Half of the sentences (half of approach and half avoidance) were set as same sentences and the other half were changed sentences (see Table 1). Sentences were randomly presented. Participants were told to press the P key if the sentence was a same sentence and the Q key if the sentence had changed. Each sentence remained on the computer screen until a response was made, and then a new sentence was displayed. Once the memory task was completed, participants received feedback on the percentage of correct responses reached. They were thanked for their cooperation, and a short explanation of the experimental procedure was given to them for debriefing. Response recordings and stimuli presentation were controlled by E-Prime 2.0 software (Psychology Software Tools, Pittsburgh, PA, USA).

### 2.2. Results

We assume normal distribution of memory performance. The Kolmogorov–Smirnov test supported a normal distribution of the overall memory performance in the sample, *p* > 0.05. We carried out an ANOVA on percentage of correct responses with direction (approach vs. avoidance) and sentence (same vs. changed) as within-subjects factors. The main effect of sentence was significant, *F*(1, 19) = 6.35, *p* < 0.021, *η*^2^ = 0.251, performance was greater in same sentences (M = 59.50, SD = 10.62) than in changed sentences (M = 48.97, SD = 11.38). Likewise, the main effect of direction was significant, *F*(1, 19) = 10.64, *p* < 0.004, *η*^2^ = 0.359, performance was greater in avoidance sentences (M = 57.92, SD = 7.77) than in approach sentences (M = 50.55, SD = 7.64). The interaction direction x sentence did not result significant.

As can be seen, performance was greater for same sentences. Changed sentences would be more difficult to the extent that accurate responses would require a more careful and effortful encoding of character names and their roles, in the memorization phase. So, mistaken responses would be more probable. On the other hand, better performance in avoidance suggests certain precedence for memorizing avoidance in comparison to approach. This agrees with previous research that shows precedence for processing negative stimuli, such as a negativity bias [34,35]. Moreover, a better performance of avoidance might be related to arousal. Arousing words have been shown to cause better word memorization [36,37], and avoidance relationship sentences have been demonstrated to be more arousing than approach ones (see Marrero et al [21]).

We also used *d’* discrimination signal-noise index as a dependent variable according to signal detection theory. First, we transformed hits (changed sentences correctly identified) and false alarms (same sentences judged as changed) in z scores (see Macmillan, & Kaplan [26]). Subsequently, we subtracted false alarms z scores from hits for each participant and obtained the *d’* index (collapsed) for the whole sample for approach and avoidance sentences. As *d’* is an index that combined performance of same and changed sentences, we carried out an ANOVA on *d’* only with Direction (approach vs. avoidance) as within-subjects factors. The effect of Direction was significant, *F*(1,19)= 4.62, *p* = 0.045, *η*^2^ = 0.196, discriminability was better in avoidance (M = −0.274, SD = 0.979) than in approach sentences (M = −0.873, SD = 0.925).

## 3. Study 2—tDCS Effects on Memorization of Relationship Action-Sentences

### 3.1. Method

#### 3.1.1. Participants

Sixty-five undergraduate students (56 females, average age: 19.50, SD = 1.13), from the University of La Laguna (first year of psychology students) participated voluntarily in the experiment in exchange for course credits, and none of them participated in the Study 1. Participants were recruited as permitted by the availability of laboratory sessions with the target sample of volunteers (N = 75), who were randomly assigned to each of the three conditions. The minimum sample size was established in N = 20 in accordance with previous research on memory performance enhancement by tDCS [38]. Exclusion criteria were suffering from epilepsy (or having close relatives affected), migraines, brain damage, cardiac disease, or other psychological or medical conditions. As attendance was previously agreed, there were participants that did not attend the assigned session. Likewise, there were participants that did not meet some of the inclusion criteria and so were rejected on arrival at the lab. These circumstances produced assumable differences in the size of samples between conditions. Twenty-four participants were subjected to the anodal condition, 20 to the sham condition, and 21 to the cathodal condition.

#### 3.1.2. Approach and Avoidance Behavioral Scales (BIS/BAS)

The behavioral inhibition system (BIS) and behavioral activation system (BAS) scales were measured by the scales of Carver & White [39]. BAS measure individual sensitivity to reward, and BIS sensitivity to punishment [27]. The BAS scale was reliable in this study, *alpha* = 0.810, as was the BIS scale, *alpha* = 0.769.

#### 3.1.3. Design

A 2 × 2 × 3 factorial design was used, with direction (approach and avoidance) and sentence (same vs. changed) as within-subjects factors, and stimulation conditions of anodal, cathodal, and sham as between-subjects factors. The dependent measure was *d’* index in performance in the memory task.

#### 3.1.4. Protocol for tDCS Application

A CE-certified battery-powered stimulator (neuroConn DCSTIMULATOR. neuroConn GmbH, Albert-Einstein-Str.3, 98693 Ilmenau, Germany) was used for the non-invasive tDCS current conduction with an intensity of 2 mA. The electrodes of the equipment used were rubber, with a size of 5 × 5 cm, and covered with sponges soaked in saline to transfer direct current, which would result in a density of 0.08 mA/cm^2^. One electrode was placed on the scalp in accordance with International System 10–20. The selected area was T8, as it is the most appropriate for the stimulation of the temporal region of interest. The other electrode was placed extracranially on the contralateral shoulder, to minimize its effects on the brain. We stimulated BA 22 and BA 21 brain areas overlapping medial aspects of rSTS, close to the brain area showed by the source stimulation analysis (see Figure 1), as shown in Figure 2. In addition, the stimulated area is a part of the so-called mentalizing [2], specialized in processing social intentionality.

The stimulation application time either excitatory or inhibitory was 20 min plus a Fade in and Fade out of 15 s both. The stimulation time was established based on previous studies of tDCS (e.g., [38,41]). During the false tDCS (sham) condition, the constant current only lasted 30 s: Fade in: 15 s and Fade out: 15 s.

#### 3.1.5. tDCS Procedure

Upon arrival at the laboratory, participants were informed about the general aim of the study. They filled in a personal data form and a questionnaire to screen for exclusion conditions and signed an informed consent form. Participants were told that the objective of the study was to examine the effect of brain stimulation on performance of a memory task. They were not informed about the tDCS condition they had been submitted. None of them reported suffering from epilepsy (nor having close relatives affected), migraines, brain damage, cardiac disease, or other psychological or medical conditions. All participants were right-handed, according to the Edinburgh Handedness Inventory [42]. The ethical committee of the University of La Laguna approved the study. Participants were given the BIS/BAS scales. Subsequently, the electrodes were placed, and tDCS stimulation started in accordance with tDCS protocol. Immediately after the tDCS session, participants initiated the experimental task. In addition to debriefing, they were advised not to discuss the experiment with other potential participants. They were thanked for their cooperation, and a short explanation of the experimental procedure was given to them for debriefing. The experimental session lasted around 30 min. The stimulation parameters are considered to be safe [43]. We asked participants to inform us of any adverse events during tDCS application. We asked the subjects again about any adverse effects at the end of the experimental session and told them to let us to know whether they felt such effects in the following days. Some volunteers informed us of mild and transient adverse effects (see Brunoni et al. [44]) during intervention. Table 2 shows the type of adverse effect, the severity of the effect and the percentage of the participants that experienced them.

Response recordings and stimuli presentation were controlled by E-Prime 2.0 software (Psychology Software Tools, Pittsburgh, PA, USA).

### 3.2. Results

Mean and standard deviations of percentage of correct responses under the different conditions are shown in Table 3.

We assume normal distribution of memory performance. The Kolmogorov–Smirnov test supported a normal distribution of the overall memory performance in the sample, *p* > 0.05. We carried out an ANOVA on *d’* with stimulation (anodal, cathodal and sham (false stimulation) conditions) as a between-subjects factor and direction (approach vs. avoidance) as a within-subjects factor. Mean and standard deviations of *d’* index in the different conditions are shown in Table 4. A 0.05 level of significance was chosen.

Figure 3 shows the distribution of discriminability scores in approach and avoidance sentences in the memory task.

The main effect of stimulation was significant, *F*(2, 62) = 4.85, *p* = 0.011, *ηp*^2^ = 0.135. Anodal stimulation improved discriminability compared to sham and cathodal condition (see Figure 4). Follow-up comparisons showed significant differences in averaged discriminability between anodal-sham conditions M_diff_. = 0.61, *t*_(42)_ = 2.11, *p* = 0.041 and anodal-cathodal conditions M_diff_. = 1.03, *t*_(43)_ = 2.89, *p* = 0.006. There was no significant difference between sham and cathodal conditions. Neither the main effect of direction nor the interaction stimulation x direction were significant.

In Study 1, we found that *d’* was greater for avoidance than for approach sentences when the task is performed alone. Thus, if anodal stimulation in rSTS exerts an effect on improvement of memorization of approach sentences, we could expect an interaction direction × stimulation in the contrast between the anodal and task-alone conditions. This interaction would show that anodal stimulation causes greater improvement of discriminability for approach than for avoidance sentences.

We carried out an ANOVA to compare anodal and task-alone conditions. We found a main effect of stimulation, *F*(1, 42) = 25.37, *p* < 0.001, *η*^2^ = 0.377. Averaged *d’* was greater in anodal (M = 0.76, SD = 0.98) than in task-alone (M = −0.57, SD = 0.72) conditions. The interaction stimulation x direction was significant, *F*(1, 42) = 4.52, *p* = 0.039, *η*^2^ = 0.097. Follow-up comparisons showed that anodal stimulation caused *d’* improvement for approach (M_diff_ = 1.81), *t*_(42)_ = 5.10, *p* < 0.001, greater than for avoidance sentences (M_diff_ = 0.86), *t*_(42)_ = 2.54, *p* = 0.015. The main effect of direction did not result as significant. The comparison of task-alone with either sham or cathodal conditions did not show significant interactions.

Our results support that anodal stimulation improved discriminability compared to sham and cathodal conditions in both approach and avoidance sentences. The contrast between anodal and task-alone conditions suggests that anodal stimulation caused a greater improvement in approach than in avoidance sentences.

#### Moderation of tDCS Effect by Affective Traits

We examined modulation by affective traits of tDCS effect on *d’* index in the memory task. Modulatory analyses are aimed at examining whether or not tDCS affects memory performance of participants depending on having a more high or low level either in BAS or BIS traits. To do this, and following Marrero, Gámez, & Díaz [45], we approximately median-split the total sample (N = 64) in high and low for each affective trait—BAS (approach) and BIS (avoidance). Subsequently, we performed correlations transforming tDCS into a continuous “stimulation” variable with *d’* scores, both for low and high participants in each trait. We transformed tDCS according to the expected effect on performance—anodal (positive effect): 1, sham (no effect): 0, and cathodal (negative effect): −1. Positive significant correlations would imply an effect of stimulation on memorization. The minimal sample size (see Faul, Erdfelder, Buchner, & Lang [46]) to generate appropriate statistical power (0.80) with 0.05 alpha bilateral for a medium correlation (*r* = 0.5) was calculated at 23. One participant in the cathodal condition did not do the BIS/BAS scale. Moreover, we carried out planned t-tests comparisons in order to examine differences in discriminability between low and high trait participants associated with stimulation. Correlation between the split BAS and BIS was not significant (*r* = 0.151, *p* > 0.10), and so showed that there was no association between them. Correlation matrix and mean and standard deviations of *d’* are shown in Table 5 for low and high trait.

In the case of the BAS trait, we found significant correlations in low BAS participants (N = 30) of stimulation with averaged *d’* (*r* = 0.537, *p* = 0.002) and approach *d’* (*r* = 0.551, *p* = 0.002). Neither of the correlations reached significance in the case of high approach trait (N = 34) (see Table 5). Planned t-test comparisons showed that in anodal condition, discriminability was significantly greater in low approach (N = 13) than in high approach (N = 11) in averaged *d’*: Mdiff. = 0.87, *t*
_(22)_ = 2.37, *p* = 0.027, and *d’* approach: Mdiff. = 1.13, *t*
_(22)_ = 2.232, *p* = 0.036, but it did not differ in *d’* avoidance, *p* > 0.10. Low and high BAS participants did not differ in discriminability in sham and cathodal conditions, *p* > 0.10.

In the case of BIS trait, we found significant correlations in low BIS participants (N = 29) of stimulation with averaged *d’* (*r* = 0.556, *p* = 0.002), approach *d’* (*r* = 0.511, *p* = 0.005), and avoidance *d’* (*r* = 0.410, *p* = 0.022). Neither of the correlations reached significance in the case of high avoidance trait (N = 34) (see Table 5). Although the trend of the mean indicates greater discriminability in low (N = 10) than in high (N = 14) BIS in anodal condition (see Table 5b), planned t-test comparisons did not show significance, *p* > 0.10. The main correlations with stimulation—*d’* average score and *d’* approach for low BAS and’ *d’* average for low BIS—remain significant and is near significance for approach *d’* in low BIS participants when Bonferroni correction is applied *(p* = 0.0041).

Correlational results support that only low BAS and BIS participants benefitted from tDCS in their discriminability in the memory task, which supports a modulatory effect of the affective traits. In addition, *t*-test comparisons showed that anodal stimulation significantly benefitted discriminability in low BAS compared to high BAS participants, whereas in the case of BIS only, a tendency was shown of greater discriminability in low BIS to high BIS participants.

## 4. General Discussion

In this study, we have examined the effect of tDCS in the right (anterior to middle) temporal lobe (T8, 10/20 system) on memory of relationship action-sentences. More specifically, we hypothesized that anodal stimulation rather than sham and cathodal stimulation would improve *d’* index of discriminability in a subsequent same-changed memory task of intentional relationship-action sentences. Our results support this hypothesis. Anodal stimulation would furnish additional processing resources, which would enable a deeper sentence encoding and thus produce better memory discrimination in contrast to sham (no stimulation) or inhibitory cathodal stimulation. In terms of recognition, anodal stimulation would improve memorization of proper names as a recollection task [47,48]—that is, proper names would be encoded in the context of the sentences (as subject and object of a certain action). This encoding during memorization would facilitate subsequent discrimination of changed sentences from unaltered ones in the memory task.

We also hypothesized a greater effect of excitatory stimulation on memorization of approach sentences. In accordance with this hypothesis, the comparison between anodal and task-alone performance suggested that anodal stimulation produced a greater improvement in *d’* of approach sentences, inasmuch that, without tDCS procedure, discriminability was greater for avoidance sentences.

We have found that *d’* scores differ significantly between anodal, on the one hand, and sham and cathodal conditions, on the other, whereas sham and cathodal conditions did not differ, even though the trend of results suggests an impairment of discriminability due to inhibitory stimulation. Although the role of anodal stimulation has been supported by previous research, the effect of cathodal stimulation is less clear (see references [49,50]). There is a certain consensus that anodal stimulation, either with 1 or 2 mA of intensity, has an excitatory effect in improvement performance of cognitive tasks, including working memory tasks; for example, over the dorsolateral prefrontal cortex [49]. In contrast, the effect of cathodal stimulation is less clear [49]. Dedoncker, Brunoni, Baeken, & Vanderhasselt, [49], concluded that analyses revealed a small, significant effect of anodal tDCS, but not in cathodal tDCS on improving accuracy and latency in several tasks, and that stimulation parameters as stimulation current were not predictive of RTs after anodal tDCS. Also relevant, it has been showed that cathodal stimulation in motor cortex had an excitatory effect on cortical excitability at 2 mA of stimulation intensity [50].

Previous research on the effect of tDCS on temporal areas in memory enhancement support excitatory effect of anodal stimulation associated with semantic processing [38,51,52,53]. However, no clear effect of cathodal stimulation emerges [38,54]. In one study [38], the left ATL was stimulated by tDCS with 2 mA of intensity to examine recognition effects associated with processing semantic relatedness. It was shown that anodal tDCS enhanced performance at test (reduced false recognition in the case of associative lists) but cathodal tDCS did not lead to any behavioral impairment. Thus, additional research is necessary to further examine the effect of cathodal stimulation on memorization of relationship-action sentences. For example, the duration of stimulation in the region of interest could be increased.

### 4.1. tDCS Effect on Memorization is Modulated by Approach/Avoidance Trait

Approach-BAS modulated the tDCS effect on discriminability in the memory task. Correlational analysis showed that low BAS participants benefitted from anodal stimulation, whereas stimulation had no effect in high BAS participants. Moreover, t-test comparisons showed that low BAS had greater discriminability than high BAS participants in the anodal condition, particularly in approach sentences. These results agree with previous research that found a greater effect of anodal stimulation in low approach (BAS) participants [28] and support the attentional explanation, though they rule out the motivational explanation. High approach (reward sensitivity) has been associated with less concentration, more distractibility, and less attentional narrowed focus on a given task [30,31,32]. Thus, high BAS participants would be less able to take advantage of additional processing resources plausibly furnished by anodal tDCS to encoding intentional action-relationship sentences, compared to low approach ones.

Avoidance (BIS) trait also modulates the tDCS effect in discriminability in the memory task. Correlational analysis showed that low BIS participants benefitted from anodal stimulation, whereas stimulation had no significant effect on discriminability of high BIS participants. This result also supports the attentional explanation. One plausible reason is that fearfulness and anxiety disturb the capacity for allocation of additional processing resources furnished by anodal stimulation to the task (see Eysenk et al. [33]).

### 4.2. Limitations, Contributions and Future Directions

Our design is a between-subjects design. This type of design is used in tDCS cognitive enhancement research (see Fan, Mao, Jin, & Ma [55]). However, it has some limitations. For example, the groups could differ in discriminability capacity; that is, participants in the anodal group could have had better capacity than participants in sham and cathodal groups before stimulation. To control for this possibility, participants were assigned to the conditions by random from a sample of first-year psychology students. Moreover, our participants are supposed to have certain homogeneity in cognitive capacities (which includes memory), as they should all have a medium to high score in the national test for university access. Another limitation of our study is that support for recruitment of rSTS in encoding approach intentionality in relationship actions has been based on indirect evidence such as source estimation analysis of ERPs in a previous study, or the effect of tDCS stimulation on memorization in the present study. For the anatomical localization of the STS, we considered the position of electrode T8 of the EEG montage; however, aspects such as the anatomical variability across subjects and the lack of focality of the applied stimulation would have played an important role in the results. Further research is thus necessary to confirm this role of rSTS by means of techniques such as fMRI or TMS that would enable more direct and precise evidence for it. In this regard, TMS has been showed to be useful for examining the process of potentiation of consolidation of memory traces, which is of interest for research in memorization processes in general [56]. Moreover, our participants are young university students with a high percentage of females. However, approach and avoidance brain encoding could be affected by developmental changes or be modulated by gender. Thus, future studies should also include adult and more male participants.

It could be argued that what our results support is a general enhancement of memorization instead of a specific improvement of memorization of interpersonal action-sentences. In fact this alternative explanation could not be discarded inasmuch the list of sentences to be memorized lacked of control sentences (without interpersonal content of approach and avoidance). In this regard, the fact that we stimulate a brain area that forms a part of the mentalizing network (see Kennedy, & Adolphs [2]), specialized in processing social intentionality suggests a specific effect of tDCS on memorization of attitudinal action-sentences. In any case, further research is necessary to examine the specificity of memory improvement for interpersonal action sentences.

A strength of the present study is that it integrates approach/avoidance trait to examine neural aspects of memorization of relationship actions. Recent affective neuroscience research considers it to be scientifically relevant to relate brain measures of basic processes to individual differences [28,57,58].

Overall, and in accordance with previous research on social perception of communicative intentions [7,8,9,10,11,12], and also with the Heider and Simmel animation task [14,15], our results support the involvement of rSTS in processing social intentionality. In addition, for the first time, rSTS involvement is shown in encoding and memorization of approach/avoidance intentional relationship-actions. In our study, tDCS stimulation was over middle aspects of rSTS, as more anterior aspects of temporal lobes are involved in a more abstract processing of information [20,38,59], and intentionality [14,15,16,17,60]. According to [16], activation of more anterior aspects of STS could represent the process of giving a social relational meaning to individual actions, and approach and avoidance would constitute a basic frame or semantic category to give that meaning.

A careful encoding of intentional direction toward other individuals in relationship actions is clearly necessary for efficient social navigation (see Marrero et al. [23])—to forget who is a friend, or to confuse friends with hostile others, has relevant consequences for an individual’s survival and thriving [61,62]. Moreover, friendship or hostility would depend on relationship actions of past interactions that should be both encoded and memorized. However, our results suggest that more impulsive (high BAS) and anxious (high BIS) participants were unable to use additional resources furnished by tDCS to memorize intentional relationship-action sentences. Further research is thus necessary to examine the reason for this inability, and whether it is associated with attentional bias or deficits in encoding intentional relationship-actions. Likewise, it could be of interest to examine deficits in encoding direction of relationship actions, in particular, in clinical syndromes associated with the deterioration of more anterior aspects of temporal lobes—for example, in semantic dementia or frontotemporal dementia [63], which have been shown to be associated with interpersonal deficits such as loss of knowledge about people’s names and faces.

## 5. Conclusions

The rSTS is involved in processing action intentionality from the posterior aspect in social perception to more anterior to middle aspects in more conceptual semantic processing of social intentionality. On the other hand, language describes how individuals interact with other people by means of verbal expressions of relationship-actions that conceptually involve approach and avoidance. Thus, approach and avoidance would constitute a semantic frame or category to be systematically encoded for understanding this type of actions. Thus, we would expect that tDCS stimulation of middle aspects of rSTS improves recollection of relationship-action sentences and then discriminability in a same-changed memory task. Our results support this prediction. Moreover, they suggest a greater effect of anodal stimulation in approach sentences, which suggests specialization of STS in processing approach intentionality in social relationship with linguistic materials, beyond social perception. Importantly, we found that anodal effect on improvement of discriminability was modulated by BAS and BIS traits. These findings are relevant to brain research on the mentalizing network for social relationship action understanding.

## Figures and Tables

**Figure 1 brainsci-10-00497-f001:**
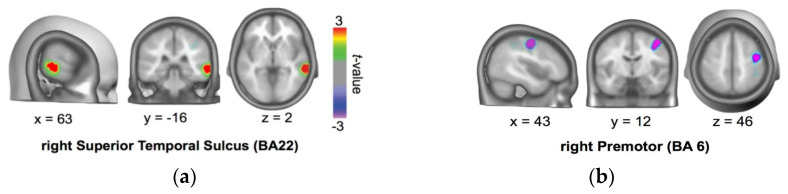
Source localization of approach/avoidance difference in the critical window ERP (545–750 ms). Source localization indicates stronger activations for approach than avoidance at the anterior/middle right STS (BA21 and 22) (**a**). Smaller activations for approach than avoidance were shown at right middle frontal gyrus (BA6) (**b**) [23].

**Figure 2 brainsci-10-00497-f002:**
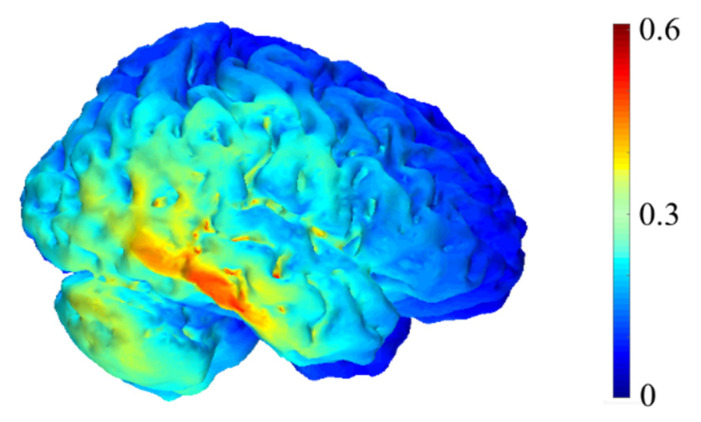
Computational representation of the electric field intensity generated by our transcranial direct current stimulation (tDCS) montage with reference to anode (T8) and an extracephalic cathode. Units are in V/m. The simulation was run using COMETS2 [40].

**Figure 3 brainsci-10-00497-f003:**
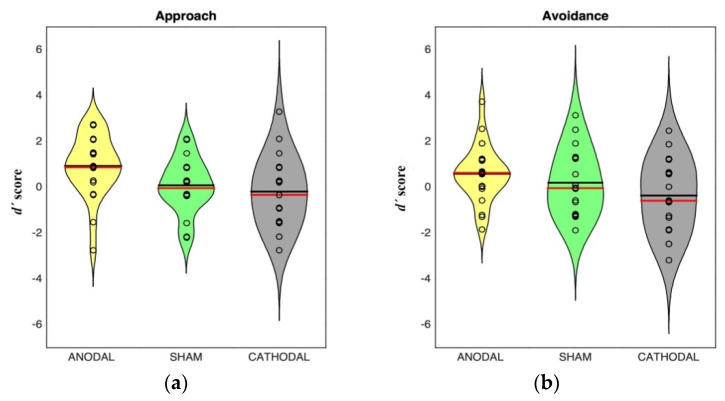
Distribution of discriminability scores in approach (**a**) and avoidance (**b**) sentences in the memory task for each stimulation group. Shaded areas represent scores probability density estimated with a kernel density estimator. Dark lines indicate mean, red lines indicate median.

**Figure 4 brainsci-10-00497-f004:**
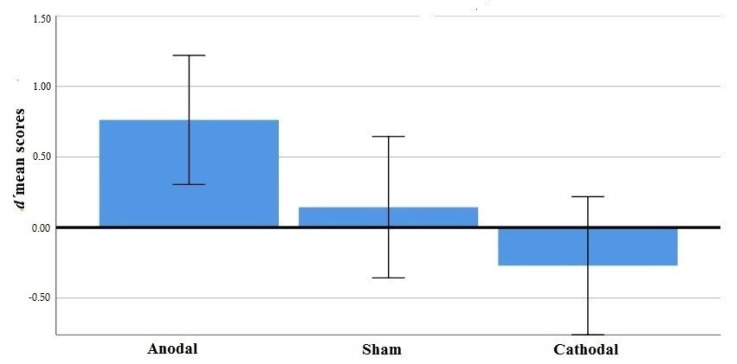
*d*’ mean scores and standard errors in tDCS conditions.

**Table 1 brainsci-10-00497-t001:** Examples of sentences in the memorization phase and the memory task (with approximate translation into English).

Memorization	Approach/Avoidance List	Same-Changed Judgement Task	Same/Changed Sentence
Pedro admitió a Rosa en el Whatsapp. (Pedro accepted Rosa in Whatsapp.)	Approach list 1	Pedro admitió a Rosa en el Whatsapp. (Pedro accepted Rosa in Whatsapp.)	Same
Pedro bloqueó a Rosa en el Whatsapp. (Pedro blocked Rosa on Whatsapp.)	Avoidance list 2	Rosa bloqueó a Pedro en el Whatsapp. (Rosa blocked Pedro onWhatsapp.)	Changed
Inés rechazó a Roberto por su papel. (Ines rejected Roberto for his role.)	Avoidance list 2	Inés rechazó a Roberto por su papel. (Inés rejected Roberto for his role.)	Same
Inés elogió a Roberto por su papel. (Inés praised Roberto for his role.)	Approach list 1	Roberto elogió a Inés por su papel. (Roberto praised Inés for his role.)	Changed

**Table 2 brainsci-10-00497-t002:** Adverse effects, severity, and percentage of participants that experienced them in the tDCS study.

Type of Effect	Severity	Percentage
Tingling	Mild	29.23%
Itching	Mild	20.00%
Sleepiness	Mild	12.30%

**Table 3 brainsci-10-00497-t003:** Means of rounded percentages of correct responses and standard deviations (within parenthesis) for the memory task in the tDCS conditions as a function of direction and type of sentence.

Sentence	Same				Changed				
Direction (tDCS)	Approach		Avoidance		Approach		Avoidance		N
Anodal	71	(15.69)	73	(13.98)	60	(17.56)	56	(16.89)	24
Sham	70	(12.76)	70	(16.22)	48	(14.09)	53	(17.80)	20
Cathodal	62	(18.33)	62	(13.64)	50	(14.30)	52	(17.00)	21

**Table 4 brainsci-10-00497-t004:** Means of *d’*scores and standard deviations (within parenthesis) for the memory task in the tDCS conditions as a function of Direction.

Direction	Approach	Avoidance	Averaged
Anodal	0.936 (1.34)	0.587 (1.21)	0.761 (0.985)
Sham	0.087 (1.18)	0.200 (1.40)	0.144 (0.941)
Cathodal	−0.182 (1.52)	−0.358 (1.58)	−0.270 (1.391)

**Table 5 brainsci-10-00497-t005:** Pearson correlation of transformed continuous tDCS variable with high and low behavioral inhibition system (BIS) and behavioral activation system (BAS) traits. (**a**). Means of *d’*scores and standard deviations (within parenthesis) for the memory task in anodal condition for low and high BIS and BAS traits (**b**).

	(**a**) Pearson Corr. tDCS with d’				(**b**) ANODAL			
BIS/BAS Trait	*d’* Approach	*d’* Avoidance	*d’* Averaged	N	*d’* Approach	*d’* Avoidance	*d’* Averaged	N
Low BAS	0.551 (*p* = 0.002)	0.341 (*p* = 0.065)	0.537 (*p* = 0.002)	30	1.45 (1.01)	0.87 (1.29)	1.16 (0.84)	13
High BAS	0.096 (*p* > 0.20)	0.136 (*p* > 0.20)	0.132 (*p* > 0.20)	34	0.32 (1.46)	0.25 (1.07)	0.28 (0.95)	11
Low BIS	0.511 (*p* = 0.005)	0.410 (*p* = 0.027)	0.556 (*p* = 0.002)	29	1.30 (1.21)	0.64 (.94)	0.97 (0.83)	10
High BIS	0.163 (*p* > 0.20)	0.123 (*p* > 0.20)	0.169 (*p* > 0.20)	35	0.67 (1.40)	0.54 (1.41)	0.60 (1.40)	14

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
