# Peer review of "Enhancing Memory for Relationship Actions by Transcranial Direct Current Stimulation of the Superior Temporal Sulcus"

_brainsci, 2020, doi:10.3390/brainsci10080497_

Round 1

Reviewer 1 Report

For the main study (study 2) which followed a behavioral experimental task pilot study (study 1), the authors compared 3 independent samples in a between-subject manner, using a 2 x 2 design of a memory task (same/difference) employing approach/avoidance verbal material. Samples differed in the application of transcranial direct current stimulation (tDCS) over the right superior temporal sulcus (rSTS), which was sought to influence the behavior on the above-mentioned stimulus material and furthermore interact with  – or rather be modulated by – behavioral traits (approach/inhibition) as measured with the BIS/BAS scales.

The manuscript is well written and structured in a way that it is easy to follow. I only have two minor comment and would therefor, and without hesitation, suggest publication.

Minor comments:

  1. A total of 12 Pearson correlations between BIS/BAS values (high/low) and approach/avoidance/averaged d‘ task performance has been calculated. As these correlations are not entirely independent, and one should consider a multiple comparisons correction. Now, my little investigation into this topic didn’t result in any definite solution, so I would suggest to at least include a sentence that even with correction for multiple comparisons using the Bonferroni correction, the main correlations (d‘ average score and d‘ approach for low BAS) remain significant and are just shy of significance for approach low BIS. That would not change the overall results or interpretation, but does make the manuscript a little bit more thorough.
  2. The numbers for the correlation statistics in the text (lines 382 and 383) do not match the numbers in the table – please correct either one (whichever does not contain the valid results).

Author Response

  1. A total of 12 Pearson correlations between BIS/BAS values (high/low) and approach/avoidance/averaged d‘ task performance has been calculated. As these correlations are not entirely independent, and one should consider a multiple comparisons correction. Now, my little investigation into this topic didn’t result in any definite solution, so I would suggest to at least include a sentence that even with correction for multiple comparisons using the Bonferroni correction, the main correlations (d‘ average score and d‘ approach for low BAS) remain significant and are just shy of significance for approach low BIS. That would not change the overall results or interpretation, but does make the manuscript a little bit more thorough.

R: Thanks for this comment. We have added the following paragraph in correlational analysis as you have suggested: "The main correlations with stimulation: d‘ average score and d‘ approach for low BAS, and d´ average for low BIS remain significant, and is near of significance for approach in low BIS participants when Bonferroni correction is applied (p = .0041)."

  1. The numbers for the correlation statistics in the text (lines 382 and 383) do not match the numbers in the table – please correct either one (whichever does not contain the valid results)

R: Done.

Reviewer 2 Report

The authors give an informed introduction of the role of the STS in social interactions and information processing. However they then mention, line 117, “On the other hand, temporal lobes are mainly involved in memory processes” without references. Although temporal lobes are associated with memory, most common medial temporal lobe regions include the perirhinal cortex, entorhinal cortex, parahippocampal cortex, hippocampus, and amygdala, and not the STS. It would thus be good if the authors can cite some work linking the STS to memory processes directly, or indirectly through connections between STS and medial temporal lobe structures. In fact instead of writing that the authors examine “the effect of tDCS stimulation in this brain region on memorization” (lines 121 and 122), it seems more accurate to write that the authors are examining the effect of STS-targeted tDCS on processing of relationship-action sentences quantified through recognition memory accuracy.

Can the authors explain why they decided to set-up the memory task such that the characters, or proper names, changed as opposed to changing the relationship-action from approach to avoidance. For example, why not after memorizing “Pedro accepted Rosa into the Whatsapp” ask participants to judge whether “Pedro blocked Rosa into the Whatsapp” was the same or different (as opposed to “Rosa accepted Pedro into the Whatsapp”). What the authors did appears to focus on the character and less on the action. I assume this is intentional, but am failing to see why this was done given the intended focus on approach vs avoidance relationship-actions.

On line 238-240, the authors write “We also used d´ discrimination signal-noise index as a dependent variable according to Signal Detection Theory. First, we transformed hits (changed sentences correctly identified) and false alarms (same sentences judged as changed) in z scores. I’m curious why changed sentence correctly identified were not counted as hits and, similarly, changed sentences judged as same as false alarms as these are all separate possibilities right? (These could also be classified according to the four possible outcomes: hits (changed sentence correctly identified), misses (changed sentence identified as same), false alarms (same sentence identified as changed), and correct rejections (same sentence correctly identified).

Why did the authors employ an offline tDCS study design as opposed to administering the tDCS during the experimental task?

Author Response

-The authors give an informed introduction of the role of the STS in social interactions and information processing. However they then mention, line 117, “On the other hand, temporal lobes are mainly involved in memory processes” without references. Although temporal lobes are associated with memory, most common medial temporal lobe regions include the perirhinal cortex, entorhinal cortex, parahippocampal cortex, hippocampus, and amygdala, and not the STS. It would thus be good if the authors can cite some work linking the STS to memory processes directly, or indirectly through connections between STS and medial temporal lobe structures. In fact instead of writing that the authors examine “the effect of tDCS stimulation in this brain region on memorization” (lines 121 and 122), it seems more accurate to write that the authors are examining the effect of STS-targeted tDCS on processing of relationship-action sentences quantified through recognition memory accuracy.

R: Thanks for this comment. We agree: STS is involved in processing and encoding social intentionality but it is not a brain area typically associated with memory. In order to make this clear, we removed the sentence “On the other hand, temporal lobes are mainly involved in memory processes” in the manuscript, and replaced the sentence ” With this aim, we examine the effect of tDCS stimulation in this brain region on memorization” by “With this aim, we examine the effect of STS-targeted tDCS on processing of relationship-action sentences quantified through recognition memory accuracy.”, as you have suggested.

-Can the authors explain why they decided to set-up the memory task such that the characters, or proper names, changed as opposed to changing the relationship-action from approach to avoidance. For example, why not after memorizing “Pedro accepted Rosa into the Whatsapp” ask participants to judge whether “Pedro blocked Rosa into the Whatsapp” was the same or different (as opposed to “Rosa accepted Pedro into the Whatsapp”). What the authors did appears to focus on the character and less on the action. I assume this is intentional, but am failing to see why this was done given the intended focus on approach vs avoidance relationship-actions.

R: Thanks for this comment. The suggested manipulation is relevant. We decided to change proper names due to, theoretically speaking, approach and avoidance are directional from an individual to other individual and thus encoding the subject and object of relationship actions is a part of approach /avoidance processing. Both have to be processed for a recollection task where proper names would be encoded in the context of the sentences (Ranganath, 2010), as subject and object of a certain action. We will take into account the suggested manipulation in future research.

- On line 238-240, the authors write “We also used d´ discrimination signal-noise index as a dependent variable according to Signal Detection Theory. First, we transformed hits (changed sentences correctly identified) and false alarms (same sentences judged as changed) in z scores. I’m curious why changed sentence correctly identified were not counted as hits and, similarly, changed sentences judged as same as false alarms as these are all separate possibilities right? (These could also be classified according to the four possible outcomes: hits (changed sentence correctly identified), misses (changed sentence identified as same), false alarms (same sentence identified as changed), and correct rejections (same sentence correctly identified).

R: You are right, there are these four possibilities as you mentioned. According to Macmillan & Kaplan (1985), could be computed by taking into account only two possibilities: hits and false alarms that was the procedure we used.

-Why did the authors employ an offline tDCS study design as opposed to administering the tDCS during the experimental task?

R: This offline tDCS procedure is usual in cognitive improvement by tDCS research (see Díez, E., Gómez-Ariza, C. J., Díez-Álonso, A., Alonso, M. A., Fernandez, A., 2017. The processing of semantic relatedness in the brain: Evidence from associative and categorical false recognition effects following transcranial direct current stimulation of the left anterior temporal lobe. Cortex, 93, 133-145). In addition, we consider appropriate to differentiate stimulation (before) from memorization in terms of causal interpretation of results.

Reviewer 3 Report

This paper studies the effect of tDCS on memorization of intentional relationships as encoded in sentences with a so called approach/avoidance attribute. The study appears interesting however there are issues that need to be clarified:

Comments:

-I would rather use the term acceptance /avoidance instead of “approach/avoidance” specially within the context of language.

-In the abstract there are abbreviations (BIS/BAS) that need to be introduced when first presented, e.g. BIS/BAS (line 16).

-Although participants have apparently equivalent memory skills. It is important to control for the memory level of participants.

Why were most of the participants included in the study are females? Please clarify.  Perhaps the title should reflect this aspect.

-Please specify the inclusion criteria for participation.

-How much time lasted a typical experiment?

-How did the authors control the drowsiness of participants? How did the authors control for the level of attention?

The following sentence is unclear: Line 196-198 : “Sentence presentation was segmented as in the following example: “Pedro/ bloqueó/ a/ Rosa/ en el/ Whatsapp” (“Pedro blocked Rosa in Whatsapp”). ”

-Table 1: the memorization phase the examples presented are only males performing an action, please also include an example that includes a female.

-Please provide as supplementary material the list of all the sentences used in the behavioral task.

-Is the experimental paradigm program available to other researchers? If so please provide the code.

Please provide graphs of behavioral percentage change between before and after stimulation, across trials and subjects.

How was the intensity of tDCS stimulation selected? How did the authors decide about the stimulation time (20 min.)?

-For the anatomical localization of the STS the authors considered the position of electrode T8 of the EEG montage, however aspects such as the anatomical variability across subjects and the lack of focality of the applied stimulation would have played an important role in the results. Please state this critical limitation.

-As indicated in lines 449-451 techniques such as TMS are useful to clarify the role of some brain areas in memory. It would be good to discuss about recent studies highlight the mechanism of memory formation by using TMS, see (Human Depotentiation following Induction of Spike Timing Dependent Plasticity. Biomedicines 2018, 6, 71.

Author Response

Comments:

-I would rather use the term acceptance /avoidance instead of “approach/avoidance” specially within the context of language.

R: Thanks for this suggestion. We agree that acceptance would be linguistically better. However, approach would have the advantage of relating the topic with previous research on brain processing of social intentionality in the mentalizing network. 

-In the abstract there are abbreviations (BIS/BAS) that need to be introduced when first presented, e.g. BIS/BAS (line 16).

R: Done

-Although participants have apparently equivalent memory skills. It is important to control for the memory level of participants.

R: We agree. Random distribution of participants to tDCS conditions is a form of control, if we take into account that university students have to be relatively homogenous in memory skills. However, the lack of a control of memory skills is a limitation of our Study 2, and it is pointed out in the Discussion: “Limitations, contributions and future directions”.

-Why were most of the participants included in the study are females? Please clarify.  Perhaps the title should reflect this aspect.

R: The percentage of female students is high in Psychology degree in our university, and our sample was of convenience.  This gender unbalance has been pointed out in the manuscript as limitation of our Study 2 in the Discussion: “Limitations, contributions and future directions”

 -Please specify the inclusion criteria for participation.

R: Thanks for this comment. Now we have added the following paragraph in Participants section of Study 2: “Exclusion criteria were: to suffer from epilepsy (nor having close relatives affected), migraines, brain damage, cardiac disease, or other psychological or medical condition”.  

 -How much time lasted a typical experiment?

R: In Study 2 around 30 minutes. We have added a sentence in the manuscript in Procedure “The experimental session lasted around 30 minutes”.

 -How did the authors control the drowsiness of participants? How did the authors control for the level of attention?

R: Participants were told in the instructions that they would receive feedback on the percentage of correct responses they reached in the task of recall. This would be an incentive to them to pay attention during the memorization phase. In addition, the experimental session in Study 2 lasted around 30 minutes that it seems not long enough to produce drowsiness.  We have added the following sentence in the Procedure of Study 1: “They were also told that they would receive feedback on the percentage of correct responses they reached in the memory task”.

 -The following sentence is unclear: Line 196-198 : “Sentence presentation was segmented as in the following example: “Pedro/ bloqueó/ a/ Rosa/ en el/ Whatsapp” (“Pedro blocked Rosa in Whatsapp”). ”

R: We have changed the sentence to make it more clear: “Sentence presentation was segmented (six segments); for example: “Pedro/ bloqueó/ a/ Rosa/ en el/ Whatsapp” (“Pedro/ blocked/ Rosa/ at the/ Whatsapp”).

 -Table 1: the memorization phase the examples presented are only males performing an action, please also include an example that includes a female.

R: Done. We have replaced the second example in Table 1with an example of a woman performing an action (in yellow).

Memorization

Same-changed judgement task

Pedro admitió a Rosa en el Whatsapp

(Pedro accepted Rosa into the Whatsapp)

Approach-list 1

Pedro admitió a Rosa en el Whatsapp

(Pedro accepted Rosa into the Whatsapp)

Same

Pedro bloqueó a Rosa en el Whatsapp

(Pedro blocked Rosa at the Whatsapp)

Avoidance-list 2

Rosa bloqueó a Pedro en el Whatsapp

(Rosa blocked Pedro at the Whatsapp)

Changed

 Inés rechazó a Roberto por su papel

(Ines rejected Roberto for his role)

Avoidance-list 2

     Inés rechazó a Roberto por su papel

(Inés rejected Roberto for his role)

Same

 Inés elogió a Roberto por su papel

(Inés praised Roberto for his role)

Approach-list 1

      Roberto elogió a Inés por su papel

(Roberto praised Inés for his role)

Changed

 -Please provide as supplementary material the list of all the sentences used in the behavioral task.

R: Done.

 -Is the experimental paradigm program available to other researchers? If so please provide the code.

R: The experimental paradigm is not available. We would give it by request. 

 -Please provide graphs of behavioral percentage change between before and after stimulation, across trials and subjects.

R: There is no measure of memory performance before stimulation in our Study 2. In a between-subjects design participants firstly memorized sentences and subsequently performed the memory task, randomly assigned to tDCS conditions. Lack of a previous measure of memory performance has been pointed out as a limitation of our study at the beginnig of  4.2. Limitations, contributions and future directions (lines 442-449).

 -How was the intensity of tDCS stimulation selected? How did the authors decide about the stimulation time (20 min.)?

R: We adopted it of other pieces of research on memory enhancement by tDCS stimulation (see Díez, E., Gómez-Ariza, C. J., Díez-Álonso, A., Alonso, M. A., Fernandez, A., 2017. The processing of semantic relatedness in the brain: Evidence from associative and categorical false recognition effects following transcranial direct current stimulation of the left anterior temporal lobe. Cortex, 93, 133-145.

 -For the anatomical localization of the STS the authors considered the position of electrode T8 of the EEG montage, however aspects such as the anatomical variability across subjects and the lack of focality of the applied stimulation would have played an important role in the results. Please state this critical limitation.

R: We agree. We have added the following paragraph as a limitation in the manuscript “Limitations, contributions and future directions”: For the anatomical localization of the STS we considered the position of electrode T8 of the EEG montage, however aspects such as the anatomical variability across subjects and the lack of focality of the applied stimulation would have played an important role in the results….”.

Also relevant, the computational representation of the Electric Field Intensity generated by our tDCS montage with reference to anode (T8) and an extracephalic cathode (Figure 2) suggests stimulation in the selected ROI.

-As indicated in lines 449-451 techniques such as TMS are useful to clarify the role of some brain areas in memory. It would be good to discuss about recent studies highlight the mechanism of memory formation by using TMS, see (Human Depotentiation following Induction of Spike Timing Dependent Plasticity. Biomedicines 20186, 71.

R: We agree and have added this paragraph: “…In this regard, TMS has been showed to be useful for examining the process of potentiation of consolidation of memory traces, which is of interest for research in memorization processes in general (Pedroarena-Leal, Heidemeyer, Trenado, & Ruge, 2018).”